# Hippocampal Cytokine Release in Experimental Epileptogenesis—A Longitudinal In Vivo Microdialysis Study

**DOI:** 10.3390/brainsci12050677

**Published:** 2022-05-21

**Authors:** Kai Siebenbrodt, Vanessa Schütz, Lara S. Costard, Valentin Neubert, Daniel Alvarez-Fischer, Kerstin Seidel, Bernd Schmeck, Sven G. Meuth, Felix Rosenow, Sebastian Bauer

**Affiliations:** 1Epilepsy Center Frankfurt Rhine-Main, Department of Neurology, University Hospital Frankfurt and LOEWE Center for Personalized Translational Epilepsy Research (CePTER), Goethe-University Frankfurt, 60528 Frankfurt am Main, Germany; vanessa.schuetz@kgu.de (V.S.); rosenow@med.uni-frankfurt.de (F.R.); sebastianbauermz@gmx.de (S.B.); 2Department of Neurology, Philipps University, 35037 Marburg, Germany; laracostard@rcsi.ie (L.S.C.); valentin.neubert@uni-rostock.de (V.N.); 3Tissue Engineering Research Group, RCSI Department of Anatomy and Regenerative Medicine, Royal College of Surgeons in Ireland, D02 YN77 Dublin, Ireland; 4Oscar-Langendorff-Institute of Physiology, Rostock University Medical Center, 18057 Rostock, Germany; 5Institute of Neurogenetics, University of Lübeck, 23538 Lübeck, Germany; daniel.alvarez@neuro.uni-luebeck.de; 6Institute for Lung Research, Universities of Giessen and Marburg Lung Center, Philipps-University Marburg, 35037 Marburg, Germany; kseidel@bu.edu (K.S.); bernd.schmeck@uni-marburg.de (B.S.); 7German Centers for Lung Research (DZL) and for Infectious Disease Research (DZIF), SYNMIKRO Centre for Synthetic Microbiology, Philipps-University Marburg, 35037 Marburg, Germany; 8Department of Neurology, Medical Faculty, Heinrich Heine University Düsseldorf, 40225 Düsseldorf, Germany; meuth@uni-duesseldorf.de

**Keywords:** epilepsy, temporal lobe epilepsy, inflammation, hippocampal sclerosis, rat model

## Abstract

Background: Inflammation, particularly cytokine release, contributes to epileptogenesis by influencing the cerebral tissue remodeling and neuronal excitability that occurs after a precipitating epileptogenic insult. While several cytokines have been explored in this process, release kinetics are less well investigated. Determining the time course of cytokine release in the epileptogenic zone is necessary for precisely timed preventive or therapeutic anti-inflammatory interventions. Methods: Hippocampal extracellular levels of six cytokines and chemokines (IL-1β, IL-6, IL-10, CCL2, CCL3, and CCL5) were quantified at various time points during epileptogenesis in a rat model of mesial temporal lobe epilepsy with hippocampal sclerosis (mTLE-HS) using microdialysis (MD). Results: The analysis of microdialysates demonstrated consistent elevation at all time points during epileptogenesis for IL-1β and IL-10. IL-10 release was maximal on day 1, IL-1β release peaked at day 8. No correlation between local hippocampal IL-1β concentrations and IL-1β blood levels was found. Conclusion: The release kinetics of IL-1β are consistent with its established pro-epileptogenic properties, while the kinetics of IL-10 suggest a counter-regulatory effect. This proof-of-concept study demonstrates the feasibility of intraindividual longitudinal monitoring of hippocampal molecular inflammatory processes via repetitive MD over several weeks and sheds light on the kinetics of hippocampal cytokine release during epileptogenesis.

## 1. Introduction

Mesial temporal lobe epilepsy with hippocampal sclerosis (mTLE-HS) is one of the most common focal epilepsies. Treatment is often challenging, with a significant frequency of pharmacoresistance, as well as cognitive and psychiatric comorbidity. Rather than targeted therapeutic strategies, the mainstay of treatment is symptomatic inhibition of seizures using antiseizure medications [1] or resective epilepsy surgery [2].

The exact underlying mechanism of mTLE-HS pathogenesis has yet to be established. It is known that an initial precipitating injury (IPI), such as complex febrile seizures, traumatic brain injury or stroke can initiate epileptogenesis and thereby manifest epilepsy [3]. Epileptogenesis is a clinically silent period—patients are asymptomatic during this time. Currently, no reliable biofluid biomarkers of epileptogenesis are established, hampering epileptogenesis research in human clinical trials.

Emerging evidence indicates that inflammatory processes increase seizure susceptibility and are involved in epileptogenesis after an IPI [4]. Immune-mediated pathogenesis has been claimed not only for autoimmune epilepsies (such as Rasmussen or antibody-mediated autoimmune encephalitis), but also for epilepsy due to specific lesions like hippocampal sclerosis (HS) in particular [5]. Several inflammatory mediators contribute to the activation of those immune responses, such as cytokines, interferons, growth factors and tumor necrosis factors [6]. 

In the central nervous system (CNS), cytokines and chemokines are primarily released by infiltrating immune cells, such as granulocytes, macrophages and lymphocytes, as well as other cell types (e.g., endothelial cells). Studies have shown an important contribution of glial cells as well as neurons in the production of cytokines during episodes of inflammation in the brain [7]. They are usually released in a cascade as one cytokine causes its target cells to produce other cytokines. Some cytokines act as precipitants of both pro-inflammatory and anti-inflammatory pathways. Interleukin-1β (IL-1β), IL-6 and the chemokines CCL2 (Monocyte Chemotattractant Protein 1 [MCP-1]), CCL3 (Macrophage Inflammatory Protein 1-Alpha [MIP-1α]) and CCL5 (RANTES) are considered pro-inflammatory and IL-10 mediates mainly anti-inflammatory processes by inhibition of pro-inflammatory cytokines and cell proliferation [8]. Besides acting as inflammatory mediators, increasing evidence indicates that cytokines also interact with receptors and ion channels regulating neuronal excitability and synaptic plasticity, as well as tissue remodeling under both physiological and pathological conditions [9]. The most convincing evidence supporting the involvement of cytokines in epileptogenesis and epilepsy progression originated from studies on IL-1β, its corresponding receptor (IL-1R1) and its antagonist (IL-1RA). Most of these study findings are based on histopathological intracellular findings [4]. The correlation between blood cytokine levels and intracerebral cytokine concentrations within the epileptogenic cortex is unknown. Intracerebral microdialysis (MD), with its probe acting like an artificial blood capillary, measures the concentration of the components of the extracellular space and is a tool broadly used in clinical practice and research [10]. 

In this study, epileptogenesis was induced using electrical stimulation of the perforant pathway (PPS) in rats, as described in [11]. This model is characterized by histopathological findings resembling the most common human HS (classified after the international league against epilepsy (ILAE)) type 1, with cell loss mostly in CA1 and CA4, and spontaneous seizure occurrence after a silent period of several days to weeks, making it well-suited for studies on epileptogenesis [12]. 

This proof-of-concept study investigates:The feasibility of longitudinal monitoring of cerebral molecular inflammatory processes using hippocampal MD over several months in an animal model of mTLE-HS (proof-of-concept);Whether and to what extend the cytokines IL-1β, IL-6, IL-10, CCL2, CCL3 and CCL5 are secreted in the hippocampus;The intraindividual release kinetics before, during and after epileptogenesis;If the MD procedure itself leads to the release of cytokines and chemokines.

## 2. Materials and Methods

### 2.1. Animals

This study and all procedures on animals were performed in accordance with the ARRIVE guidelines [13] and with the guidelines of the European Community (Directive 2010/63/EU). All experiments were approved by the local regulatory authority (Regierungspräsidium Gießen). Twenty male Sprague Dawley rats weighing from 250 to 300 g were housed in groups of two to six in standard type 4 cages with a 12 h light/12 h dark cycle with access to water and food ad libitum for at least seven days prior to surgery. Animals were then equally assigned to either the experimental or control group; no randomization was performed to allocate subjects in the study.

### 2.2. Surgery

A detailed description of the surgical procedure for this animal model without cerebral microdialysis has been given in a previous paper [14]. In short, rats were anaesthetized with inhalative isoflurane (1–4% Forane^®^ and 100% oxygen, 0.6 L/min) and pain was treated with buprenorphine (0.2 mg/kg s.c.). Each head was fixed in a stereotactic frame and the skull was exposed by removing the skin and periosteum. Holes (1 mm in diameter) were drilled into the skull to bilaterally implant stimulation electrodes into the perforant pathway (4.5 mm lateral to bregma, 0.5 mm rostral to lambda, variable depth) and a recording electrode was implanted into the left hippocampus (2 mm lateral, 3 mm caudal, variable depth). A guide cannula for MD probes (CMA Microdialysis AB, Kista, Sweden) was inserted above the right hippocampus (5.2 mm caudal of bregma, 4.6 mm lateral right, 3.5 mm depth). The position of the electrodes is shown in Figure 1. One additional drill hole was placed rostral to the bregma; two additional holes were drilled between the lambda and the bregma for a transmitter ground screw and an additional support screw. A wireless EEG transmitter (OpenSource Instruments [OSI], Waltham, MA, USA) was implanted subcutaneously on the right flank, allowing ground and recording cables to reach the skull via a subcutaneous tunnel.

The final positioning of the electrodes was determined by the maximal amplitude and shape of the recorded population spikes when stimulating the perforant pathway during surgery. Electrodes and the MD guide cannula were fixated with dental acrylic cement and a small cap with a pedestal was formed. After surgery, rats were given a seven-day recovery period.

### 2.3. Induction of Epileptogenesis and Microdialysis

The perforant pathway was stimulated on three consecutive days (stimulation for 30 min on day 1 and day 2 and 8 h on day 3; pulse duration 0.1 ms, frequency 2 Hz, voltage 20 V, twin pulses with an interpulse interval of 40 ms and additional trains of 20 Hz single pulses at 20 V applied once per minute for 10 s) to induce mTLE-HS [11,14].

MD in the epilepsy group was performed before the first stimulation (baseline) and additional MDs were performed on days 1, 8 and 15 after PPS being time points during epileptogenesis as well as 30 days after the first spontaneous seizure. The control group had the same MD time points and PPS took place only in the epilepsy group. For an annotated timeline, see Figure 2. Prior to MD, 500 µL of blood was sampled via the tail vein and centrifuged for 10 min at 5000 rpm. The MD probe (high cut-off 100 kDa, CMA 12, CMA Microdialysis AB, Kista, Sweden) was lowered into the hippocampus through the guide cannula. Using an automatic syringe pump, a flow rate of 1 µL of artificial cerebrospinal fluid (aCSF: NaCl 58.44 g/mol, KCl 74.56 g/mol, CaCl2 147.02 g/mol, MgCl2 203.3 g/mol, Glucose 180.16 g/mol, 0.5% BSA in distilled water, pH 7.2 to 7.4, sterile filtered) per minute allowed the collection of 60 µL of sample per hour in the awake animal, which was free to roam, eat and groom in its cage during the entire procedure. The first 60 µL were discarded to allow the establishment of a steady state, as recommended by the manufacturer. In total, 4 × 60 µL samples per dialysis were collected and to each sample, 1 µL of protease inhibitor (Creative Enzymes, Shirley, NY, USA) was added. The obtained serum and MD samples were stored at −80 °C.

Apart from not receiving PPS, the control animals were treated the same as the epileptic animals. After the final MD, animals were deeply anaesthetised, first with inhalative isoflurane (5% Forane^®^), then using an intraperitoneal application of Xylazine (15 mg/kg) and Ketamine (100 mg/kg) and transcardially perfused with 0.9% saline and 4% paraformaldehyde.

### 2.4. Video EEG Monitoring

The subcutaneous EEG transmitter (Open Source Instruments [OSI] Inc., Waltham, MA, USA, 0.3–160 Hz, sampling rate 512 samples per second, sixteen-bit sample conversion factors 0.41 microvolts per count, 3.0 mL, 3200 hr operating life) allowed wireless EEG recording using the OSI telemetry system to record single channel hippocampal local field potentials from the recovery phase until the end of the experiment. Infrared camera recordings were used to obtain additional video surveillance. Recordings were digitally stored and visually examined. All events that clearly differed from baseline activity were rated by at least two reviewers with high levels of experience in EEG analysis for seizure detection and additionally correlated with the semiology gained from the video. 

### 2.5. Cytokine Concentration Measurements

Concentrations of IL-1β, IL-6, IL-10, CCL2, CCL3 and CCL5 in MD samples were measured in the first and fourth 60 µL samples of each dialysis with a magnetic bead-based assay (Luminex Magpix, RECYTMAG-65K, Merck Milliplex, Schwalbach, Germany). Each sample was divided into two parts for duplicate analysis. Serum levels of IL-1β were measured by sandwich ELISA (Pelikine, Sanquin, Amsterdam, The Netherlands). 

### 2.6. Statistical Analysis

For depiction of the hippocampal cytokine release, the values of the first hour (h1) were averaged with the values of the last hour (h4) and the results are shown as scatter plots with the mean value annotated. To evaluate the effect size between the groups, we calculated Cohen’s d with a value of d > 0.8 considered a large effect size. P-values for mean value differences between the groups were calculated using the *t*-test for two independent samples; differences between baseline and the different time points were calculated with a paired samples *t*-test. The correlation coefficients for comparison of different MD time points (h1 and h4) were calculated using the Pearson’s coefficient. Spearman’s rank correlation coefficient was calculated between the cytokine levels in blood and MD for each animal.

Statistical analysis and graphic presentation were performed using Microsoft Excel 2003 (or later versions), SPSS Statistics (version 28.0.1.0, IBM, Armonk, NY, USA) and GraphPad Prism 9 (GraphPad Prism version 9.0.0 for Windows, GraphPad Software, San Diego, CA, USA, www.graphpad.com accessed on 18 April 2022).

## 3. Results

Twenty rats underwent surgery as described above, with seven animals excluded due to death during periprocedural anaesthesia (*n* = 5) or cap loss (*n* = 2). The exclusions were unrelated to the group affiliation. The epilepsy group included seven animals and the control group six. 

After PPS, acute symptomatic seizures occurred in six of the seven animals of the epilepsy group and the mean duration until the first spontaneous seizure was 21.7 days (range of 5–61 days). The mean seizure frequency of spontaneous seizures was 2.8 seizures/week (range of 1.5–4.6 seizures/week). One animal did not develop spontaneous seizures despite receiving PPS and having initial acute symptomatic seizures at the time of PPS. This animal has been excluded from the analysis so that six animals from each group went into the final analysis. The longest EEG follow up was 118 days, in the animal that did not develop spontaneous seizures. The longest period between the first and last MD was 117 days.

### 3.1. Hippocampal Cytokine Release

Compared to the baseline, IL-1β and IL-10 showed consistent elevations at all time points during epileptogenesis (d1, d8, d15). Maximum IL-10 release in the epileptogenesis group occurred on day one after PPS (20.0 ± 11.7 pg/mL at d1 vs. 9.7 ± 6.6 pg/mL at baseline, *p* = 0.040; 15.8 ± 10.4 pg/mL at d8 vs. 9.7 ± 6.6 pg/mL at baseline, *p* = 0.025). The maximum IL-1β-release compared to the baseline in the epileptogenesis group was seen on day eight (mean value 2.6 ± 2.0 pg/mL on d8 vs. 0.7 ± 0.8 pg/mL at baseline, *p* = 0.075) and in control animals, the concentrations showed no considerable changes during the time course and especially no elevations compared to the baseline. By comparing these augmented values in the epilepsy group with the ones of the control group at the respective time points, we see a distinct elevation in the epilepsy group for IL-10 and IL-1β at all time points during epileptogenesis, with the clearest difference for IL-10 on d1 (20.0 ± 11.7 pg/mL in epileptic animals vs. 6.7 ± 8.7 pg/mL in controls, *p* = 0.049, d = 1.29) and IL-1β on d8 (2.6 ± 2.1 pg/mL in epileptic animals vs. 0.7 ± 1.0 pg/mL in controls, *p* = 0.069, d = 1.18). Cohen’s d suggests a large effect size between the groups for both cytokines. In the MD at the last time point with established epilepsy, the cytokine levels in the epileptic animals converged with the ones in the control group (Figure 3). The other cytokines and chemokines demonstrated no changes of note, but augmented values of IL-6 and CCL2 were observed in both groups at the final time point (Figure 4).

### 3.2. Release Kinetics during the MD Procedure

To determine the time course of the values during the MD procedure itself, the value of the first collected sample (h1) in controls and the epilepsy group with the last collected sample (h4) were compared. For every examined parameter, there was a positive correlation coefficient, regardless of the group affiliation. For IL-1β, as well as the chemokines CCL2, CCL3 and CCL5, the values were significantly higher in the later sample (*p* < 0.05; Table 1).

### 3.3. Comparison of IL-1β-Levels in Blood and MD

Blood results were compared to the mean values of the first hour of MD (h1) and the fourth hour (h4). A comparison of IL-1β levels in the blood and microdialysates of the epilepsy group demonstrated partly-opposed time courses. The correlation coefficients varied from 0.22 to 0.6,2 with a lowest p-value of *p* = 0.21, therefore showing no correlation. For example, a strong increase in the blood level of IL-1β on d8 in animal #13 was not reflected in MD values. The results are shown in Figure 5.

## 4. Discussion

This study served as proof-of-concept that hippocampal MD is a suitable method for longitudinal monitoring of extracellular inflammatory processes and the measurement of hippocampal cytokine release kinetics during epileptogenesis in awake, freely moving rats. For this model of mTLE-HS, this was the first longitudinal MD study of epileptogenesis. Repetitive MD allows for longitudinal monitoring in individual animals with comparatively low disruption to the wellbeing of the animals, contributing to reduction and refinement in accordance with the 3R principle. MD reflects the composition of extracellular fluid, allowing the measurement of location-specific secretory processes that play a critical role in inflammatory processes. A large body of evidence from existing studies is based on tissue evaluation showing intracellular or membrane-bound molecules. Thus, MD is a useful and relevant supplement for histological studies. Previously, Meller et al. examined hippocampal concentrations of acetylcholine and amino acid neurotransmitters via MD in rats using a pilocarpine model and electrical stimulation of the basolateral amygdala during a period of 10 weeks, which also demonstrates the feasibility of long-term cerebral microdialysis [16]. 

### 4.1. Secretion of Cytokines during Epileptogenesis

#### 4.1.1. IL-1β 

This study provides additional evidence for the active secretion of IL-1β and IL-10 in early epileptogenesis. The data support IL-1β being crucially involved in epileptogenesis, as shown in several clinical and experimental studies [17]. IL-1β is assigned a key role in the orchestration of the complex immune response to infection and injury. It is released by macrophages and other infiltrating immune cells, but also synthesized in the brain by glial cells and certain neurons [18]. IL-1 receptors are found in different regions of the CNS with the highest abundance in the hippocampus [19]. Overexpression of IL-1β, IL-1R1 and IL-1RA is demonstrated in the rodent brains of different animal models after electrically or chemically induced SE [20]. In this study, maximal IL-1β-release occurred on day 8 and converged to the baseline after established epilepsy. Expression of IL-1β has been shown to peak 6 h after electrically induced SE in rats, with the subsequent inflammatory response lasting several weeks [6]. IL-1β expression in glial cells remains elevated for up to 60 days [21]. In addition, IL-1β influences neuronal function under pathological and physiological conditions [19], with direct application of IL-1β into the CNS increasing neuronal excitability and amplifying bicuculline- and kainic acid (KA)-induced seizures [22]; this also lowers the seizure threshold in experimental models of febrile seizures [23]. 

In human brain tissue resected during epilepsy surgery, elevated expression of IL-1β and IL-1RA was found in focal cortical dysplasia (FCD) and glioneuronal tumors [24], as well as in cortical tubers and giant cell astrocytoma in patients with tuberous sclerosis [25] and in hippocampal sclerosis [26]. CSF samples of children with epilepsy showed significantly elevated IL-1β values compared to controls [27]. 

The pro-convulsive and pro-epileptogenic effects of IL-1β have been associated with amplified glutamatergic neurotransmission, with enhanced excitotoxicity via IL-1R1-induced activation of the N-metyhl-D-Aspartate (NMDA) receptor [28]. Furthermore, IL-1β-induced transcriptional activations of MAP-Kinase and NF-κB are thought to induce morphological and physiological changes in neuronal networks, contributing to epileptogenesis [29].

Based on these findings, antagonization of IL-1β may be a therapeutic target for the prevention of epileptogenesis. Pharmacological blockade of IL-1β/IL-1R1-axis, using the IL-1 antagonist Anakinra, reduced the duration and frequency of seizures in rodents after induction of SE via pilocarpine or electrical stimulation and exerted neuroprotective effects, including reduced cell loss in the frontal brain regions [30,31]. There are case reports that suggest Anakinra could be effective in controlling epileptic activity [32]. Of note, the data of this study imply that therapeutic intervention should take place soon after the IPI. 

#### 4.1.2. IL-10 

IL-10 is broadly expressed by many immune cells in both the adaptive (e.g., Th1, Th2 and Th17 cell subsets, regulatory T cells, CD8+ T cells and (regulatory) B cells) as well as the innate immune system (e.g., dendritic cells, macrophages and neutrophils) [8,33,34]. Induction of IL-10 often occurs together with pro-inflammatory cytokines, although pathways that induce IL-10 may negatively regulate these pro-inflammatory cytokines [34]. In this context and consistent with its role in the periphery, activation of the IL-10 receptor in the CNS leads to decreased release of TNF, IL-1β, IL-6, IL-8, IL-12, and IL-23; reduced proliferation of T cells and decreased antigen presentation of monocytes and macrophages [35]. IL-10 has the capacity to act in a neuroprotective manner via the inhibition of NF-κB, thereby reducing neurotoxic accumulation of glutamate. IL-10 is therefore a functional antagonist to the proinflammatory IL-1β [33,35]. This study demonstrates that early secretion of IL-10, even earlier than IL-1β, into the extracellular space potentially acts to temper the inflammatory process occurring after the IPI. Of note, patients with hippocampal sclerosis have significantly reduced IL-10 serum levels, it was suggested that these patients have an inadequate anti-inflammatory immune response [36]. Another study showed upregulation of IL-10 in the resected hippocampi of epilepsy patients in the areas CA1 and CA3 [37]. Therefore, IL-10-agonism could be a further potential therapeutic strategy to prevent epileptogenesis. Various attempts to positively influence autoimmune diseases, such as multiple sclerosis, by administration of IL-10 did not demonstrate a clear effect [33]. 

#### 4.1.3. IL-6

IL-6, a potent inducer of the acute phase response, is secreted by T and B cells, macrophages, microglia, and non-immune cells such as endothelial cells or neurons [38]. In contrast, its receptor is found only on a few cell types, including some leukocytes and microglia. IL-6 plays prominent roles in chronic inflammation, autoimmunity, infectious disease and cancer and has context-dependent pro- and anti-inflammatory properties. In the pathogenesis of epilepsy, IL-6 has demonstrated a dichotomous role. Generally, IL-6 is regarded as a proinflammatory cytokine. Patients with epilepsy have shown elevated blood levels of IL-6 both interictally and postictally [39,40,41], as well as elevated CSF levels of IL-6 [42]. Overexpression of IL-6 seems to increase gliosis in the hippocampus and intranasal application of IL-6 in mice or transgenic IL-6 expression in the rodent brain has a proconvulsive effect [43]. Studies have also shown neuroprotective effects. IL-6 knockout mice showed higher susceptibility to proconvulsive stimuli in KA-induced seizures [44]. Studies investigating stroke demonstrate that IL-6 has neuroprotective effects via the regulation of oxidative stress and angiogenesis [43]. In this study, IL-6 showed no relevant increase in the hippocampal extracellular compartment in epileptogenesis. 

#### 4.1.4. Chemokines

The chemokines CCL2, CCL3 and CCL5 demonstrated no notable changes in our study. The literature suggests that chemokines, particularly CCL2, may well play a role in the pathogenesis of epilepsy. Increased levels of CCL2 are described in the resected tissue of epilepsy patients [45]. In addition, rat hippocampal concentrations of CCL2 and its receptor CCR2 were elevated following pilocarpine-induced seizures [46]. Finally, patients with mTLE have demonstrated elevated hippocampal levels of CCL3 [47]. 

### 4.2. Influence of MD and Probe Placement on Cytokine Levels

The seven-day recuperation period after surgery appears to be sufficient for recovery, as all baseline cytokine levels in both groups were relatively low. The MD procedure itself may precipitate inflammation, as comparing the first samples to those collected three hours later demonstrated that the later samples had increased cytokine levels. The placement of the MD probes provoking minor trauma in the tissue probably accounts for this cytokine increase. Earlier studies show similar results: Folkersma et al., for example, discuss cytokine release following probe placement. They also demonstrate that a steady state for extraction is reached after about 1.5 h [48]. 

### 4.3. Cytokines as Blood Biomarker in Epilepsy

Changes in the immune cell profile may serve as an indicator of differential activation of the immune system in patients with epilepsy. Changes in both the innate immune system (including monocyte subsets) and the adaptive immune system (e.g., immune patterns of lymphocyte subsets) in peripheral blood and CSF have been discussed as typical immunological features in mTLE patients [5]. In addition to circulating immune cells, plasma cytokine levels are increasingly considered as potential biomarkers in epilepsy. IL-1β, for instance, has been shown to correlate with disease severity and drug resistance [49,50]. Of note, in this study no correlation between the IL-1β concentration of the MD samples and the blood could be found. MD therefore provides crucial data that cannot otherwise be established by blood sampling alone. This finding probably implies that circulating cytokines are unlikely to be suitable biomarker candidates for the detection of epileptogenesis. 

### 4.4. Strengths and Limitations

Using in vivo MD allows for sampling of the extracellular space and is therefore a worthy addition to the well-established tissue examination protocol during epileptogenesis. This is particularly true for inflammation, wherein secretory processes play a crucial role. MD also allows longitudinal sampling, i.e., repeated measurements in the same animal at consecutive time points, thus reducing the number of animals required for experimentation. Furthermore, intra-individual monitoring provides a good opportunity for correlation with clinical parameters that change over time (e.g., seizure frequency). The rat model used here has excellent face validity for human mTLE-HS as it closely resembles the histopathology of HS ILAE type 1 and is characterized by a relatively long latency period before spontaneous seizures occur.

Limiting factors are the time-consuming and costly procedure associated with MD and indeed the specific PPS rat model. Furthermore, the small but unequivocal inflammation induced by the MD probe requires closely matched control animals. MD cannot be seen as a minimally invasive procedure [51] and delivers only small samples that require very sensitive detection methods. In addition, the MD procedure is not specific to certain hippocampal subregions and the MD probe extends over different subfields. Further studies should differentiate the exact source of the cytokines. Lastly, this study was for proof-of-concept, using a small number of animals. The results of cytokine and chemokine measurements should therefore be validated in an independent cohort.

## 5. Conclusions

Preclinical and clinical data provide evidence of a prominent role of cytokines in epileptogenesis. As cytokines are often redundant in their function, the development of novel therapies remains complicated. Thus, mechanistic studies are needed to further distinguish the signaling pathways responsible for physiological versus pathological conditions. This proof-of-concept study demonstrates that longitudinal monitoring of hippocampal cytokine release into the extracellular space is feasible over several months during epileptogenesis in freely moving rats and provides a useful, relevant supplement to histopathological studies. The release kinetics of IL-1β are consistent with its established pro-epileptogenic properties, while the kinetics of IL-10 suggest a counter-regulatory effect. These data provide a basis for sample size calculation for larger studies and for the timing of anti-inflammatory interventions in order to antagonize epileptogenesis.

## Figures and Tables

**Figure 1 brainsci-12-00677-f001:**
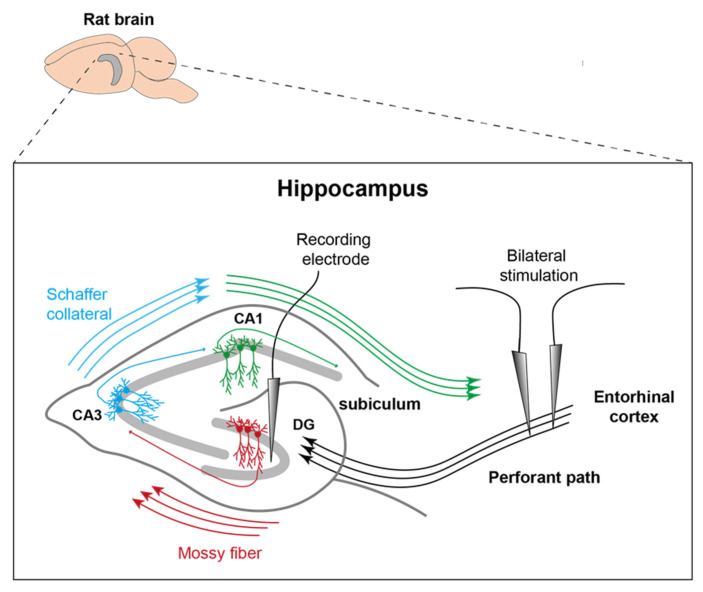
Representation of electrode position. Stimulation electrodes were inserted into the PP on both sides and the recording electrode was placed in the left DG. MD took place more posterior and along the longitudinal axis of the right posterior hippocampus (modified after [15]).

**Figure 2 brainsci-12-00677-f002:**
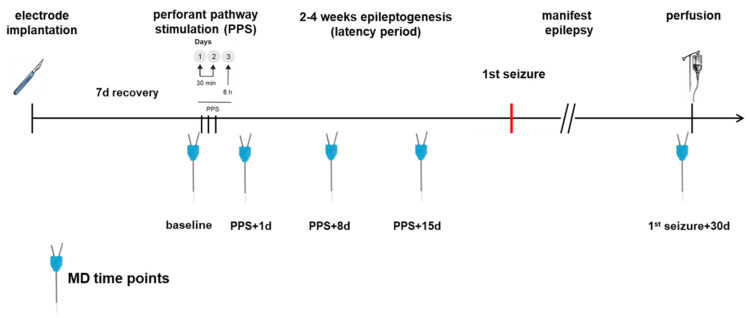
Time course of the experiment. PPS took place only in the epilepsy group.

**Figure 3 brainsci-12-00677-f003:**
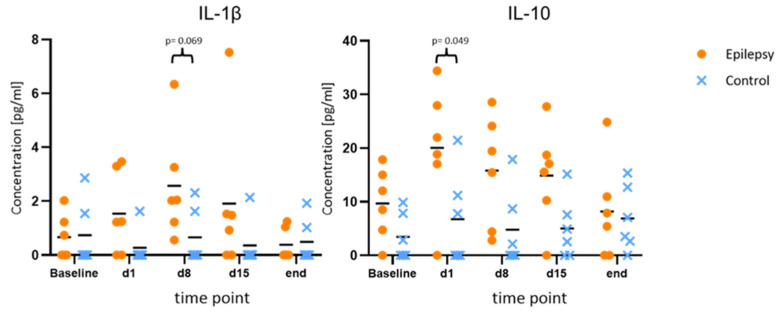
Increased values of IL-1β and IL-10 during epileptogenesis. Concentrations of IL-1β and IL-10 at baseline, during and after epileptogenesis in epileptic animals vs. controls (*n* = 6 each group) shown as scatterplots of single values with mean values [pg/mL]. The values indicate consistent elevations for both cytokines at all time points during epileptogenesis when compared to the baseline values in the epilepsy group, as well as in comparison between the groups.

**Figure 4 brainsci-12-00677-f004:**
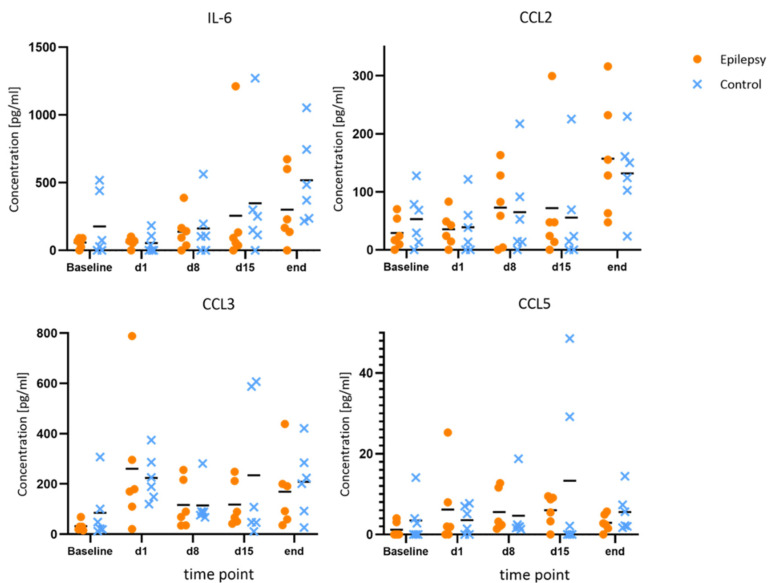
No changes of note for IL-6, CCL2, CCL3 and CCL5 during epileptogenesis. Concentrations of IL-6, CCL2, CCL3 and CCL5 at baseline, during and after epileptogenesis in epileptic animals vs. controls (*n* = 6 each group) shown as scatterplots of single values with mean values [pg/mL]. No systematic changes were found.

**Figure 5 brainsci-12-00677-f005:**
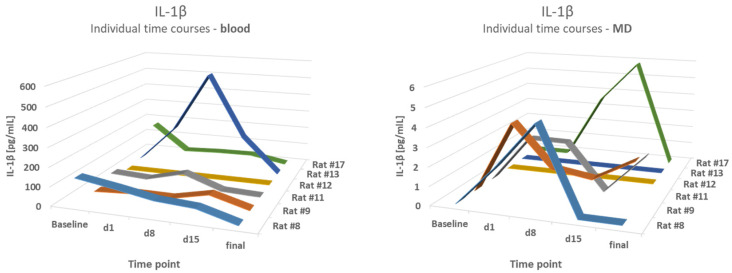
No correlation between blood and MD levels. A comparison of IL-1β levels in the blood and microdialysates (MD) of the epilepsy group at different time points.

**Table 1 brainsci-12-00677-t001:** Comparison of mean values of the first hour of MD (h1) with the fourth hour (h4). Pearson’s correlation coefficient was calculated showing a significant positive correlation of the values for IL-1β, CCL2, CCL3 and CCL5 (*n* = 12, *p* < 0.05).

Cytokine/Chemokine	Mean Value ^1^ ± SD h1	Mean Value ^1^ ± SD h4	Pearson Correlation	*p* (T ≤ t)
IL1-β	0.6 ± 0.7	1.2 ± 0.9	0.75	0.01
IL-6	130.1 ± 76.6	384.7 ± 389.3	0.29	0.057
IL-10	8.0 ± 6.6	10.5 ± 4.7	0.62	0.16
CCL2	40.5 ± 33.4	126 ± 95.7	0.23	0.016
CCL3	148.4 ± 71.6	302.1 ± 102.2	0.34	0.0008
CCL5	0.8 ± 0.7	9.4 ± 5.4	0.89	0.0002

^1^ Mean values in [pg/mL].

## Data Availability

The data presented in this study are available on request from the corresponding author.

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
