# Peer review of "Hippocampal Cytokine Release in Experimental Epileptogenesis—A Longitudinal In Vivo Microdialysis Study"

_brainsci, 2022, doi:10.3390/brainsci12050677_

Round 1
Reviewer 1 Report
We quantified hippocampal extracellular levels of six cytokines and chemokines points during epileptogenesis in an animal model of mesial temporal lobe epilepsy with hippocampal sclerosis using microdialysis. They demonstrated consistent elevation of IL-1β and IL-10 at all time points during epileptogenesis and the realease of IL-10 and IL-1β was maximal on day 1st and 8th, respectively. The results provide a basis for sample size calculation for larger studies and for the timing of anti-inflammatory interventions in order to antagonize epileptogenesis. The work is very well planned, performed and written, and it exhaustively achieved the set goals. It is worth emphasizing compliance with the ARRIVE guidelines.
I only have a few minor suggestions for improvement:
- The species of animal on which the research was performed is not mentioned in the title, abstract or keywords. Please correct it.
- L. 100 There are newer guidelines (the European Union Directive of 22 September 2010 (2010/63/EU)). Please correct it.
- L. 116, 120 and others When stating the manufacturer, please mention the name of the city in addition to the name of the producer and the country.
- L. 244 Please change comma to the decimal point.
Author Response
We would like to thank the reviewers for their valuable suggestions, which we addressed as indicated below.
Reviewer 1:
I only have a few minor suggestions for improvement:
The species of animal on which the research was performed is not mentioned in the title, abstract or keywords. Please correct it.
Response 1: Thank you for the indication, in the abstract this was specified and changed from “animal model” to “rat model”. Also, “rat model” was added to the key words.
- 100 There are newer guidelines (the European Union Directive of 22 September 2010 (2010/63/EU)). Please correct it.
Response 2: This was corrected and replaced by the correct Directive 2010/63/EU.
- 116, 120 and others When stating the manufacturer, please mention the name of the city in addition to the name of the producer and the country.
Response 3: The cities were added for all manufacturers.
- 244 Please change comma to the decimal point.
Response 4: Comma has been changed to a point.
Reviewer 2 Report
In this study, the authors demonstrate that longitudinal monitoring of hippocampal cytokine release into the extracellular space is feasible during epileptogenesis in freely moving rats. They detected consistent elevation for IL-1β and IL-10 during epileptogenesis and found no correlation between local hippocampal IL-1β concentrations and IL-1β blood levels.
I have only minor comments on this study:
1) All abbreviations should be spelled out when first mentioned or not used at all (e.g., line 86 - HS ILAE type 1)
2) The data in the figure and in the text do not match, please check where the errors occur (lines 208-210 and figure 3, mean values do not match and p-values do not match).
3) Table 1. Please check the sentence "A paired-samples t-test was used showing a significant positive correlation of..." Something wrong here with statistics. I recommend to revise the table and place all the statistical data for each cytokine in one row. Column 1 is cytokine name, column 2 is mean value and standard deviation at hour 1, column 3 is mean value and standard deviation at hour 4, column 4 is statistical criterion, column 5 is p-value.
4) Comparison of IL-1β-levels in blood and MD should be revised. To talk about the presence or absence of correlation, the Spearman rank correlation coefficient between cytokine content in blood and MD for each animal can be calculated. The absence of significant correlations will allow to draw a more rigorous conclusion.
Author Response
We would like to thank the reviewers for their valuable suggestions, which we addressed as indicated below.
Reviewer 2:
I have only minor comments on this study:
1) All abbreviations should be spelled out when first mentioned or not used at all (e.g., line 86 - HS ILAE type 1)
Response 1: We checked the manuscript again and spelled all abbreviations out when first mentioned.
2) The data in the figure and in the text do not match, please check where the errors occur (lines 208-210 and figure 3, mean values do not match and p-values do not match).
Response 2: The data in lines 208-210 describe mean cytokine concentrations at baseline and at the time point of the highest release within the epilepsy and control groups. They match with the black horizontal lines of the respective time points in Figure 3., The p-values in lines 208-210 refer to the comparison of the highest release time points vs. baseline within the same experimental arm (epilepsy or control, respectively). In contrast, the p values in Figure 3 refer to comparison between epilepsy and control groups; these comparisons are addressed in the text in lines 216-221.
We clarified this issue in the text by adding the group information.
3) Table 1. Please check the sentence "A paired-samples t-test was used showing a significant positive correlation of..." Something wrong here with statistics. I recommend to revise the table and place all the statistical data for each cytokine in one row. Column 1 is cytokine name, column 2 is mean value and standard deviation at hour 1, column 3 is mean value and standard deviation at hour 4, column 4 is statistical criterion, column 5 is p-value.
Response 3: Thank you for your recommendation. We revised the table according to your suggestion and adjusted the description of the table.
4) Comparison of IL-1β-levels in blood and MD should be revised. To talk about the presence or absence of correlation, the Spearman rank correlation coefficient between cytokine content in blood and MD for each animal can be calculated. The absence of significant correlations will allow to draw a more rigorous conclusion.
Response 4: We added exemplary results of correlation coefficients using the Spearman rank correlation with respective p-values. We also supplemented the additional statistical methodology in the materials and methods section.